# Extrinsic-Riveting Friction Stir Lap Welding of Al/Steel Dissimilar Materials

**DOI:** 10.3390/ma17081830

**Published:** 2024-04-16

**Authors:** Xiangchen Meng, Jiaze Gao, Yuming Xie, Tifang Huang, Jihong Dong, Xiaotian Ma, Naijie Wang, Yongxian Huang

**Affiliations:** 1State Key Laboratory of Precision Welding & Joining of Materials and Structures, Harbin Institute of Technology, Harbin 150001, China; mengxch@hit.edu.cn (X.M.); jzgao2024@163.com (J.G.); huangtifang1993@163.com (T.H.); yxhuang@hit.edu.cn (Y.H.); 2Zhengzhou Research Institute, Harbin Institute of Technology, Zhengzhou 450046, China; xiaotian_m@126.com (X.M.); wangnj9391@163.com (N.W.); 3AVIC Manufacturing Technology Institute, Beijing 100024, China

**Keywords:** extrinsic-riveting friction stir lap welding, Al/steel, mechanical riveting, metallurgical bonding, mechanical properties

## Abstract

To obtain high-quality joints of Al/steel dissimilar materials, a new extrinsic-riveting friction stir lap welding (ERFSLW) method was proposed combining the synthesis advantages of mechanical riveting and metallurgical bonding. SiC-reinforced Al matrix composite bars were placed in the prefabricated holes in Al sheets and steel sheets, arranged in a zigzag array. The bars were stirred and mixed with Al sheets under severe plastic deformation (SPD), forming composite rivets to strengthen the mechanical joining. SiC particles were uniformly dispersed in the lower part of the welding nugget zone (WNZ). The smooth transition between the SiC mixed zone and extrinsic-riveting zone (ERZ) ensured the metallurgical bonding. The maximum tensile shear load of the joints reached 7.8 kN and the maximum load of the weld per unit length was 497 N/mm. The fracture occurred at the interface between the rivets and steel sheets rather than the conventional Al/steel joining interface. Moreover, ERFSLW can prolong the service life of joints due to three fracture stages. This method can be further extended to the welding of other dissimilar materials that conform to the model of “soft/hard”.

## 1. Introduction

The background of carbon neutrality has made demands on the development of lightweight designs for rockets, high-speed trains, and vehicles [1,2]. One effective strategy to achieve this goal is the replacement of steel with Al alloys. However, the welding between Al alloys and steel faces a big challenge due to the substantial difference in physical and mechanical properties [3]. Therefore, developing new methods which can achieve the goal of the high-quality welding of Al/steel dissimilar materials is particularly important.

Solid-state welding methods such as explosive welding can avoid the shortages of fusion welding when it comes to dissimilar materials [4,5], but many of them are only suitable for the welding of plates with a large size. Friction stir welding (FSW) has been widely used in the welding of dissimilar metals because of the characteristics such as low welding temperature and large plastic deformation [6,7,8]. Many methods were proposed to improve the mechanical properties of the joints. Changing to a new experimental design or improving the welding parameters are relatively classic strategies [9,10,11]. Adding an interlayer between the Al alloys and steel can prevent direct contact between the two materials, reducing the production of hard and brittle intermetallic compounds (IMCs) [12,13,14]. Auxiliary methods promoting the material flow such as arc preheatment [15] and ultrasonic vibration [16,17,18] were also reported in FSW between Al alloys and steel. But these methods rely only on the metallurgical bonding to improve the properties. It is worth mentioning that combining welding with mechanical joining methods is another effective way to further improve the strength of the joints [19,20,21]. Huang et al. [22,23] proposed self-riveting friction stir lap welding (SRFSLW) to introduce a new concept with the assistance of mechanical riveting via prefabricated geometrical configurations. Dynamic recrystallized Al alloys flowed downwards and filled the prefabricated holes in the steel sheets, forming a self-riveting structure. However, this action resulted in the weld thinning for about 0.70 mm. The fracture occurred at the rivets, indicating that the strengthening effect is still to be improved.

In this paper, a novel method, extrinsic-riveting friction stir lap welding (ERFSLW), was proposed for obtaining Al/steel lap joints in order to achieve micro-weld thinning joints with great mechanical properties. Dissimilar lap joints between 6082-T6 Al alloy and QSTE340TM steel were adopted. The structure and mechanical properties of the ERFSLW joints were investigated in detail.

## 2. Materials and Methods

### 2.1. Materials

The 6082-T6 Al alloy sheets (Alnan Aluminium Construction Co., Ltd., Nanning, Guangxi, China) and QSTE340TM steel sheets (Baoshan Iron & Steel Co., Ltd., Shanghai, China) were selected, whose dimensions were 300 mm × 80 mm × 3 mm, and 300 mm × 80 mm × 2 mm, respectively. Schematics of the ERFSLW process and welding tool are illustrated in Figure 1. The specific procedures are as follows: (a) The Al sheet was located at the upper region and the steel sheet was lain at the lower region in order to avoid the tool abrasion since the pin directly contacted with steel. (b) Before welding, straight-through holes in a zigzag array which had a diameter of 4 mm were prefabricated in both Al and steel sheets. The total width of two rows was 7 mm. (c) SiC-reinforced 2024 Al matrix composite bars were set in the holes. The selected reason are as follows: Firstly, SiC-reinforced 2024 Al matrix composite has been proven to be an effective strengthening material. Secondly, 6082-T6 Al alloys are easily softened under the thermo-mechanical effects. The 2024 Al matrix composite can make up for the strength loss. Last but not least, the corrosion resistance between 6082-T6 and 2024 Al matrix composite are different, which is beneficial for observing the distribution of SiC particles after etching. In order to make three holes uniformly distributed on the tensile shear specimen as a cycle, the interval between adjacent holes in a row was fixed at 10.67 mm. The bars had a diameter of 3.95 mm and a length of 5 mm. The volume fraction of SiC particles, which had an average diameter of 5.5 μm in the bar, was 45%. (d) During the welding process, the welding tool advanced along the center line of holes. Then, the bars were stirred and mixed with Al sheet, forming composite rivets. The welding tool had a concave shoulder with a diameter of 16 mm and an enlarged-end pin with a length of 3 mm. The diameter of the enlarged-end was 8 mm, which can realize the coverage of bars and Al sheet. Rotational speed, tilt angle, and plunge depth were fixed at 1000 rpm, 2.5°, and 0.1 mm, respectively. Welding speed varied from 100 mm/min to 300 mm/min.

### 2.2. Analytic Methods

The metallographic specimens and tensile shear specimens of joints were prepared by wire-cut electrical discharge machining (WEDM) according to the positions and dimensions illustrated in Figure 2. The rivets on each specimen were uniformly distributed. Macrostructure and microstructure of the joints were observed by Keyence VHX-1000E optical microscope (OM, Keyence Corporation, Osaka, Japan). Microhardness tests of the joints’ cross-section were conducted using HX-1000 Vickers hardness tester (Shanghai Changfang Optical Instrument Co., Ltd., Shanghai, China) applying a test load of 200 g with a dwelling time of 10 s. Tensile shear tests were performed at the ambient temperature under a crosshead speed of 0.5 mm/min on AG-X plus testing machine (Shimadzu Corporation, Kyoto, Japan).

## 3. Results and Discussion

### 3.1. Macrostructure and Microstructure

Among the three group of parameters, when the welding speed was fixed at 100 mm/min, the formation of joints was the best. Now, we mainly analyze the joints formed by this welding speed. Figure 3 illustrates the upper and lower surfaces of joints. No obvious defects or large black spots can be observed on the upper surface, indicating that the hard Al matrix composite was sufficiently smashed and mixed with Al alloy. The gaps between the bars and the inner wall of holes were too small to be visible, which means the rivets were formed successfully. These phenomena indicate that ERFSLW achieved a satisfactory formation of joints due to the joining between the Al alloy, Al matrix composite bars, and steel.

Figure 4 shows the macrostructure of the joints’ cross-section. The interface between the Al sheet and bars was formed under severe plastic deformation (SPD). The cross-section can be divided into five zones: the welding nugget zone (WNZ), thermal-mechanically affected zone (TMAZ), heat affected zone (HAZ), base material (BM), and extrinsic-riveting zone (ERZ). The interface between two sheets was straight since the enlarged-end pin did not make contact with the steel, which can effectively avoid the pin abrasion. Importantly, the weld thinning of joints was about 0.34 mm, which was much smaller than that of SRFSLW.

Combining the macrostructure of the joints’ longitudinal-section, it is found that the SiC particles were dispersedly distributed in the lower part of the WNZ due to the severe stirring effect of the enlarged-end pin, as is observed in Figure 5. During the welding process, as for the pin, the enlarged-end can completely stir the bars, so the particles can be fully dispersed and mixed with Al alloy. However, the diameter of the other part is smaller than the total width of the two rows of bars, which caused some of the composite to not be directly affected by the pin. Therefore, the upper part of the WNZ was the SiC unmixed zone while the lower part was the SiC mixed zone. The combination of rivets and Al alloy became the key point that decided the strength of the joints. According to Figure 6, the SiC mixed zone and ERZ had a smooth transition and no defects were observed. This is because the high strain rate can improve the interface atom diffusion between the Al matrix composite and Al alloy. In the transition zone, the size of the grains was similar and the distribution of the SiC particles was uniform, reducing the stress concentration during the tensile shear process. The above synthesis effects are together conducive to improving the strength of the joints.

In addition, the original Al matrix composite usually have many micro-cracks and micro-holes. During the cooling process of pressure casting, tensile stress will be generated at the interface between the SiC particles and Al matrix because of the huge difference in their coefficient of thermal expansion (CTE). When the stress is larger than their bonding force, micro-cracks and micro-holes will occur. According to Figure 7, ERFSLW can make materials tighter, repairing the original micro-cracks and micro-holes in the bars, which is also beneficial for improving the mechanical properties of joints.

### 3.2. Microhardness Tests

Figure 8 shows the microhardness distribution of the joints’ longitudinal-section, 1 mm above the interface between two sheets. With the increase in welding speed, the distribution of hardness was more uneven. Because the stirring time of the weld per unit length was reduced, the uniformity distribution of the SiC particles decreases. The hardness distribution of the joints at 100 mm/min and 200 mm/min were more uniform, avoiding the stress concentration. Markedly, the hardness of ERZ with SiC particles was higher than that of base Al alloy. It shows that SiC particles play a role in strengthening the rivets.

### 3.3. Tensile Shear Tests

Figure 9 reveals the typical tensile shear curves and fracture morphologies of the joints. At different welding speeds, the maximum load of the joints was basically the same, which was about 7.8 kN. The maximum load per unit length was 497 N/mm. The lower the welding speed, the longer the stirring time of the weld per unit length, which led to the higher temperature. The sufficient heat input can improve the interfacial metallurgical bonding between the Al sheets and steel sheet. With the increase in welding speed and decrease in heat input, the interfacial metallurgical bonding was gradually weak, which reduced the elongation of joints. The fracture of each specimen occurred at the interface between the rivets and steel sheets. This is because the bearing area of each sheet was reduced due to the prefabricated holes. The metallurgical bonding of the rivets and Al sheet made up for the shortage, whereas the steel sheet did not. There are three stages in each tensile shear curve. Figure 10 shows the schematic of the load bearing during the tensile shear process. A, B, and C are three important positions of the joints. The minimum width of A and B is only 0.67 mm, so it is easy to shrink at the two positions. Due to the unavoidable machining errors, the actual dimensions of the two positions will be slightly different. The synchronous fracture of the two positions caused the formation of the first and second drop in the curve. When the load was larger than the bear capacity of C, the neck contraction occurred, showing a slowly descending curve. When the final fracture occurred, the third drop appeared. In general, Al/steel dissimilar joints show the characteristics of indirect fracture. The three stages during the tensile shear process is conducive to establishing an early warning mechanism for the fracture and prolonging the service life of the joints. Besides Al/steel joints, ERFSLW can be further extended to the welding of other dissimilar materials such as Al/Ti, Al/Cu, and Al/polymer, which conform to the model of “soft/hard”.

## 4. Conclusions

(1)Combining the synthesis advantages of mechanical riveting and metallurgical bonding, ERFSLW was put forward with SiC-reinforced Al matrix composite bars placed in the prefabricating holes arranged in a zigzag row, obtaining high-quality Al/steel dissimilar joints.(2)SiC-reinforced Al matrix composite bars were smashed into small particles and dispersedly distributed in the lower part of the WNZ under SPD. The smooth transition of the SiC mixed zone and ERZ is beneficial for improving the strength of the joints effectively.(3)The maximum tensile shear load of the joints was 7.8 kN and the maximum load of the weld per unit length was 497 N/mm. The three fracture stages during the tensile shear process is conducive to establishing an early warning mechanism and prolonging the service life of the joints.(4)Since ERFSLW has successfully welded Al alloys and steel, it can be further extended to the welding of other dissimilar materials such as Al/Ti, Al/Cu, and Al/polymer, which can be considered as models of “soft/hard”.

## Figures and Tables

**Figure 1 materials-17-01830-f001:**
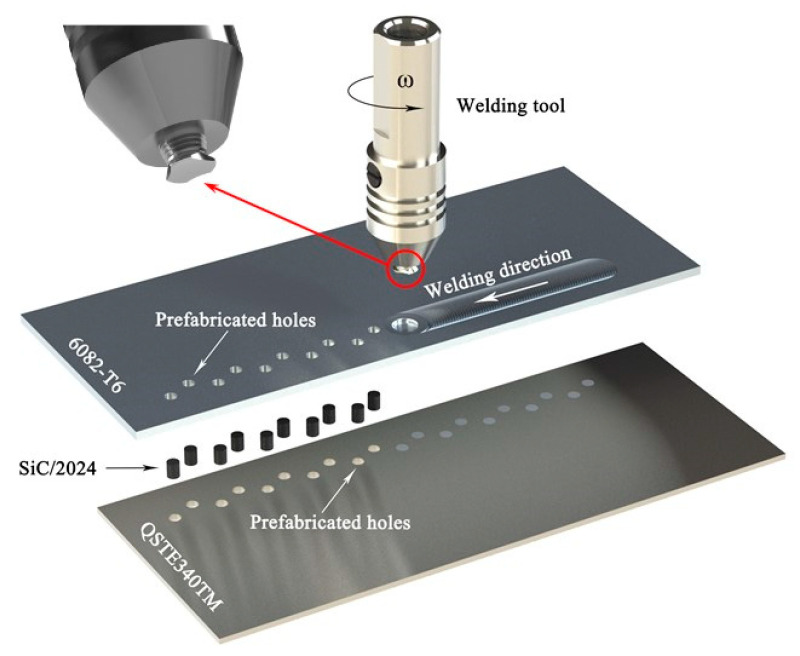
Schematics of the ERFSLW process and welding tool.

**Figure 2 materials-17-01830-f002:**
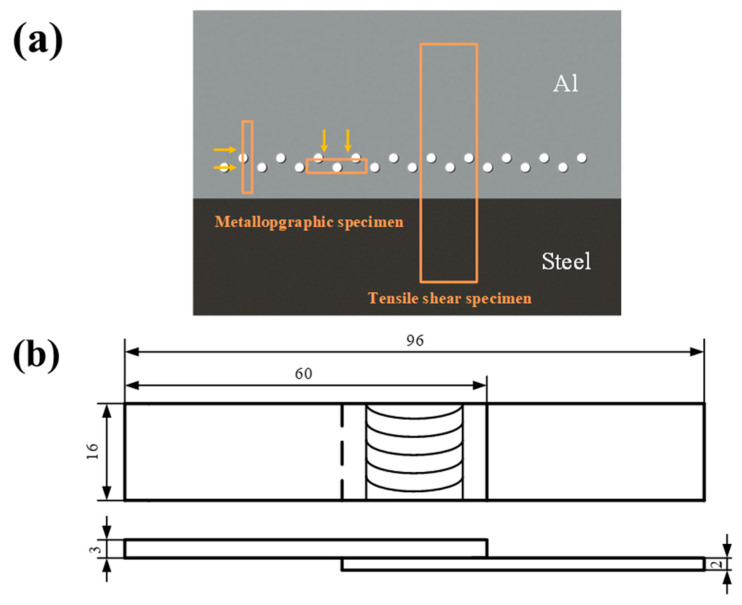
Positions and dimensions of the specimens: (**a**) positions of the specimens, and (**b**) dimensions of the tensile shear specimens (unit: mm).

**Figure 3 materials-17-01830-f003:**
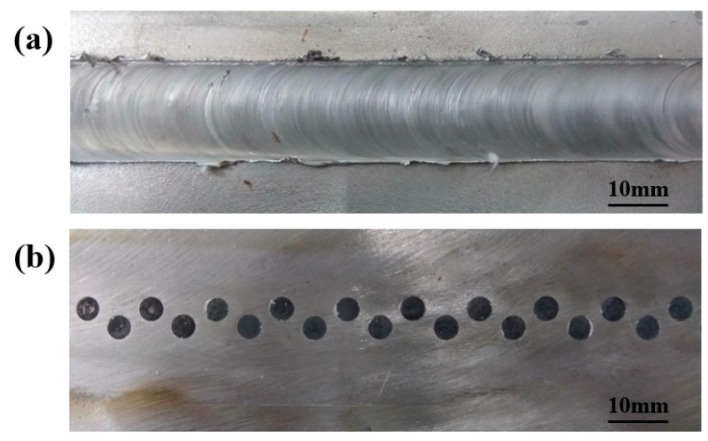
Surface formation of joints at 100 mm/min: (**a**) the upper surface, and (**b**) the lower surface.

**Figure 4 materials-17-01830-f004:**
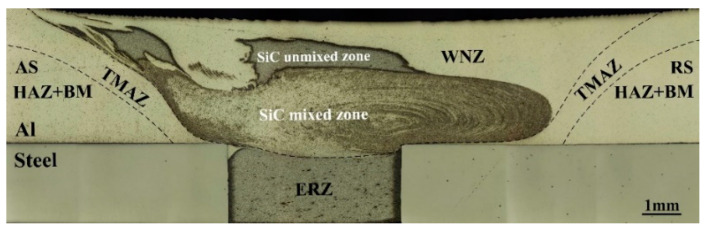
Macrostructure of the joints’ cross-section at 100 mm/min.

**Figure 5 materials-17-01830-f005:**
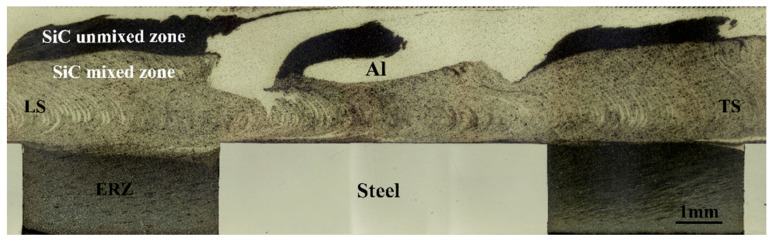
Macrostructure of the joints’ longitudinal-section at 100 mm/min.

**Figure 6 materials-17-01830-f006:**
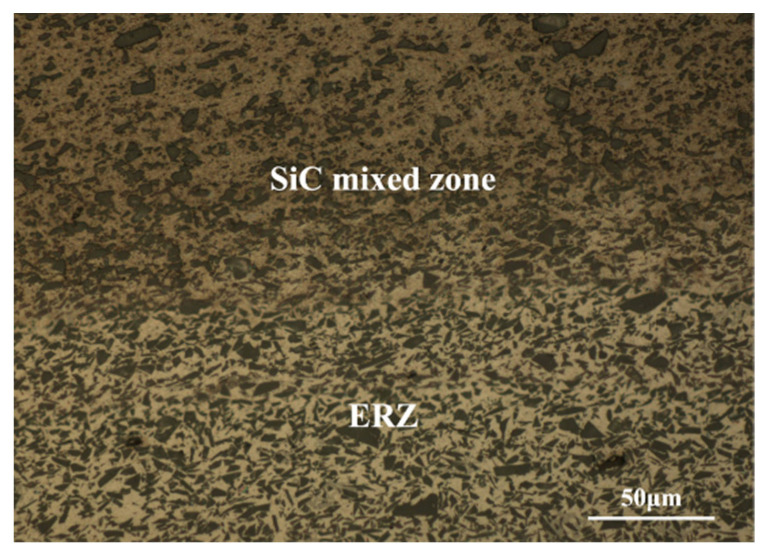
Microstructure of the transition zone between SiC mixed zone and ERZ at 100 mm/min.

**Figure 7 materials-17-01830-f007:**
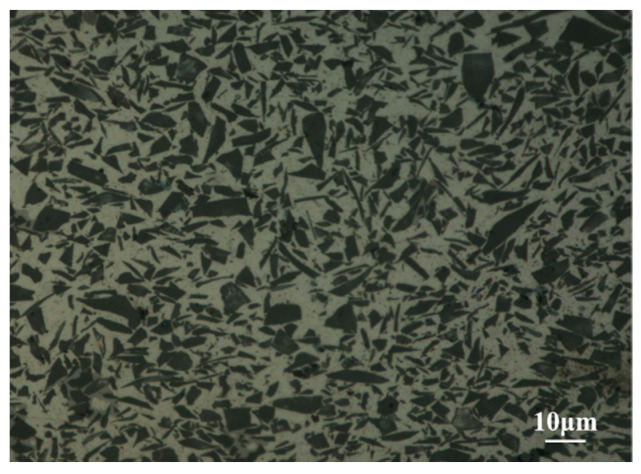
Microstructure of ERZ at 100 mm/min.

**Figure 8 materials-17-01830-f008:**
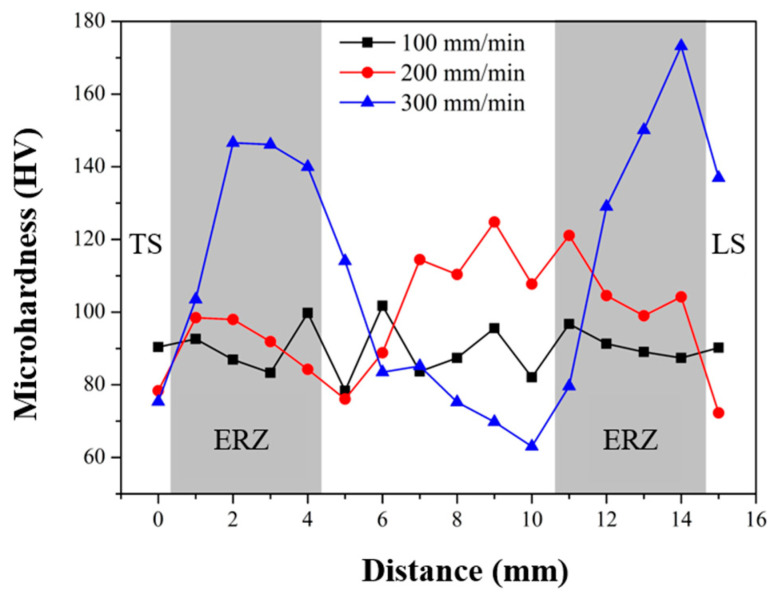
Microhardness distribution of the joints’ longitudinal-section.

**Figure 9 materials-17-01830-f009:**
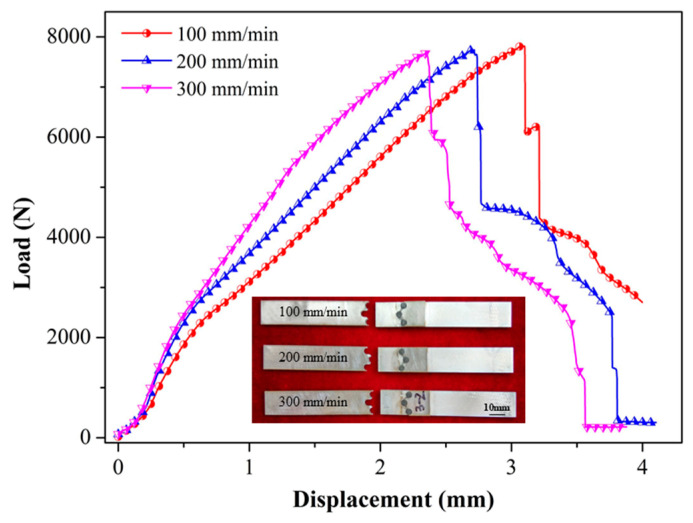
Tensile shear curves and fracture locations of the joints.

**Figure 10 materials-17-01830-f010:**
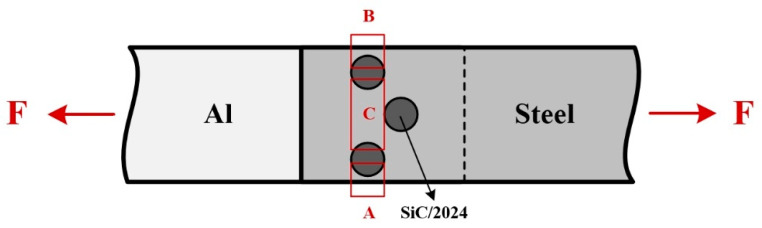
Schematic of load bearing during the tensile shear process.

## Data Availability

Data are contained within the article.

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
