# Peer review of "Extrinsic-Riveting Friction Stir Lap Welding of Al/Steel Dissimilar Materials"

_materials, 2024, doi:10.3390/ma17081830_

Round 1

Reviewer 1 Report

Comments and Suggestions for Authors

This study presents a new extrinsic-riveting friction stir lap welding (ERFSLW) method combining the synthesis advantages of mechanical riveting and welding metallurgical bonding to obtain high-quality joints of Al/steel dissimilar metals. The article is well organized. The quality of the images is good. It is recommended to apply the following before publishing.

The innovation and purpose of the research are not well stated, and the abstract and introduction should be modified and this issue should be included in it. In addition, it is mandatory to add results and achievements to the abstract.

The way of referencing in the introduction should be modified. The use of general sentences with more than four references can be seen in the introduction.

In addition, the introduction is written very briefly and superficially. It is necessary to add relevant sources and compare the process with other presented methods such as explosive welding and accumulative roll bonding and present the advantages of the mentioned method.

Use the following references to deepen the introduction. Investigation of annealing treatment on the interfacial and mechanical properties of Al5052/Cu multilayered composites subjected to ARB process. An Exhaustive Evaluation of Fracture Toughness, Microstructure, and Mechanical Characteristics of Friction Stir Welded Al6061 Alloy and Parameter Model Fitting Using Response Surface Methodology.

The percentage of similarity of the article to the literature is another challenge. This amount should be reduced to less than 20%.

In the second section, several sub-sections should be used to make the article more attractive.

How has the reproducibility of the mechanical results been checked? Add the error bar to the results.

The process of selecting welding parameters should be presented. Explain how to choose optimum welding parameters. How is the welding quality checked? Explain more about the welding process.

It is suggested to convert the force-displacement diagram into stress-strain to be comparable.

Author Response

Dear Editor/Reviewers,

Thank you for the valuable comments on the manuscript entitled Extrinsic-riveting friction stir lap welding of Al/steel dissimilar materials (materials-2947781). We really appreciate you for your carefulness and conscientiousness. Your suggestions are very valuable and helpful for revising and improving our paper. Careful modifications have been made according to the comments we received. We hope that the updated manuscript will meet the high requirements of this journal. You will find our point-by-point responses to the editor and reviewer’s comments below.

The modified manuscript has been attached, in which all the modifications are highlighted in blue. Thank you very much.

Reviewer #1:

This study presents a new extrinsic-riveting friction stir lap welding (ERFSLW) method combining the synthesis advantages of mechanical riveting and welding metallurgical bonding to obtain high-quality joints of Al/steel dissimilar metals. The article is well organized. The quality of the images is good. It is recommended to apply the following before publishing.

Comment 1: The innovation and purpose of the research are not well stated, and the abstract and introduction should be modified and this issue should be included in it. In addition, it is mandatory to add results and achievements to the abstract.

Response to Reviewer #1 comment 1: Thank you for your nice comment. The innovation and purpose have been carefully described in Abstract. We have revised as" SiC particles were uniformly dispersed in the lower part of welding nugget zone (WNZ). The smooth transition between SiC mixed zone and extrinsic-riveting zone (ERZ) ensured metallurgical bonding. The maximum tensile shear load of joints reached 7.8 kN and the maximum load of weld per unit length was 497 N/mm. The fracture occurred at the interface between the rivets and steel sheets rather than the conventional Al/steel joining interface."

Comment 2: The way of referencing in the introduction should be modified. The use of general sentences with more than four references can be seen in the introduction.

Response to Reviewer #1 comment 2: Thank you for your nice comment. The number and positions have been modified to make the manuscript better and meet requirements.

Comment 3: In addition, the introduction is written very briefly and superficially. It is necessary to add relevant sources and compare the process with other presented methods such as explosive welding and accumulative roll bonding and present the advantages of the mentioned method.

Response to Reviewer #1 comment 3: The related discussion has been added, as followed: Solid-state welding methods such as explosive welding can avoid the shortages of fusion welding when it comes to dissimilar materials [4,5], but many of them are only suitable for the welding of plates in large size.

Comment 4: Use the following references to deepen the introduction. Investigation of annealing treatment on the interfacial and mechanical properties of Al5052/Cu multilayered composites subjected to ARB process. An Exhaustive Evaluation of Fracture Toughness, Microstructure, and Mechanical Characteristics of Friction Stir Welded Al6061 Alloy and Parameter Model Fitting Using Response Surface Methodology.

Response to Reviewer #1 comment 4: The related references have been cited and discussed as [1] and [6], respectively.

Comment 5: The percentage of similarity of the article to the literature is another challenge. This amount should be reduced to less than 20%.

Response to Reviewer #1 comment 5: The manuscript has been completely revised, which has been reduced to less than 20% similarity to the literature.

Comment 6: In the second section, several sub-sections should be used to make the article more attractive.

Response to Reviewer #1 comment 6: In the second section Materials and Methods, 2.1. Materials and 2.2. Analytic Methods have been added.

Comment 7: How has the reproducibility of the mechanical results been checked? Add the error bar to the results.

Response to Reviewer #1 comment 7: In fact, we have obtained three joints with each group of parameters. The selected curves are results of typical specimens in each group.

Comment 8: The process of selecting welding parameters should be presented. Explain how to choose optimum welding parameters. How is the welding quality checked? Explain more about the welding process.

Response to Reviewer #1 comment 8: We selected the welding parameters according to the references [20], [21]and the data we have obtained by experiences.

We appreciate for Editor and Reviewers’ warm work earnestly, and hope that the revision will meet with approval. Thanks for your nice work again!

Sincerely yours,

Xiangchen Meng

Reviewer 2 Report

Comments and Suggestions for Authors

This paper well fits the scope of the journal. However, some revisions are necessary before publication.

[1] Introduction

What are differences between the “self”-riveting frictional stir lap welding (SRFSLW) in [20] and the “extrinsic”- riveting frictional stir lap welding (ERFSLW)? Please explain those in detail. The research history for SRFSLW should be described. Only one paper has been published on this welding process?

[2] 2. Materials and Methods

A reason and advantage to use SiC-reinforced 2024 Al matrix composite should be explained, even though it is popular material for improving the mechanical properties.  

[3] Line 67

“a length of 5 mm (equal to the total thickness of two sheets)” is correct? The total thickness should be 6 mm (3 mm + 3 mm).

[4] Fig. 2

The description of “The metallographic specimens” is not clear.

[5] 3. Results and discussion

It is necessary to indicate the welding speed for Figs. 3-7.

[6] Line 93

“the weld thinning of conventional SRFSLW was completely avoided.” How to evaluate this? This is the most important improvement from SRFSLW as described in the introduction. The authors must discuss it in detail.

[7] Line 151

“The fracture of each specimen occurred on the steel side instead of the rivets” Please discuss the reason.

[8] Fig. 9

A reason to cause the change in displacement in 3 conditions should be discussed.

[9] Line 180

“which can be 180 further extended to other dissimilar materials such as Al/Ti, Al/Cu, Al/polymer.” The above sentence is described in conclusion but this point is not discussed in 3. Results and discussion.

Comments on the Quality of English Language

Please check English again during the revision.

Author Response

Dear Editor/Reviewers,

Thank you for the valuable comments on the manuscript entitled Extrinsic-riveting friction stir lap welding of Al/steel dissimilar materials (materials-2947781). We really appreciate you for your carefulness and conscientiousness. Your suggestions are very valuable and helpful for revising and improving our paper. Careful modifications have been made according to the comments we received. We hope that the updated manuscript will meet the high requirements of this journal. You will find our point-by-point responses to the editor and reviewer’s comments below.

The modified manuscript has been attached, in which all the modifications are highlighted in blue. Thank you very much.

Reviewer #2:

Comment 1: Introduction. What are differences between the “self”-riveting frictional stir lap welding (SRFSLW) in [20] and the “extrinsic”- riveting frictional stir lap welding (ERFSLW)? Please explain those in detail. The research history for SRFSLW should be described. Only one paper has been published on this welding process?

Response to Reviewer #2 comment 1: Thank you for your nice comment. Huang et al. [20,21] proposed self-riveting friction stir lap welding (SRFSLW) to introduce a new concept with assistance of mechanical riveting via prefabricated geometrical configurations dynamic recrystallized Al alloys flowed downwards and filled the prefabricated holes in the steel sheets, forming self-riveting structure. However, this action easily resulted in the weld thinning for about 0.70 mm. The fracture occurred at the rivets, indicating that the strengthening effect is still to be improved. Meanwhile, Reference [21] has been added which is about SRFSLW.

Comment 2: Materials and Methods. A reason and advantage to use SiC-reinforced 2024 Al matrix composite should be explained, even though it is popular material for improving the mechanical properties.

Response to Reviewer #2 comment 2: The reason and advantages have been added and discussed, as followed: Firstly, the SiC-reinforced 2024 Al matrix composite is popular and has been proved to be an effective strengthening material. Secondly, the 6082-T6 aluminum alloys are easily softened under the thermo-mechanical effects. 2024 Al matrix composites can make up for the strength loss. Lastly, the corrosion resistance between 6082-T6 and 2024 Al matrix composite are different, which are beneficial to observing the material flow and distribution of SiC particles after etching.  

Comment 3: Line 67. “a length of 5 mm (equal to the total thickness of two sheets)” is correct? The total thickness should be 6 mm (3 mm + 3 mm).

Response to Reviewer #2 comment 3: The dimensions of base materials have been modified as “6082-T6 Al alloy sheets and QSTE340TM steel sheets were selected, whose dimensions were 300 mm × 80 mm × 3 mm and 300 mm × 80 mm × 2 mm, respectively.”

Comment 4: Fig. 2. The description of “The metallographic specimens” is not clear.

Response to Reviewer #2 comment 4: We have added “The rivets on each specimen were uniformly distributed.”. The related pictures have been revised, as follows:

Figure 2. Positions and dimensions of the specimens: (a) positions of the specimens and (b) dimensions of the tensile shear specimens (unit: mm).

Comment 5: Results and discussion. It is necessary to indicate the welding speed for Figs. 3-7.

Response to Reviewer #2 comment 5: We have added “Among three group of parameters, when the welding speed was fixed at 100 mm/min, the formation of joints was best. Now mainly analyzing the joints formed by this welding speed.”

Comment 6: Line 93. “the weld thinning of conventional SRFSLW was completely avoided.” How to evaluate this? This is the most important improvement from SRFSLW as described in the introduction. The authors must discuss it in detail.

Response to Reviewer #2 comment 6: We have contrasted the thickness of SRFSLW and ERFSLW joints according to the macrostructure. Their thickness was added as 0.34 mm and 0.70 mm, respectively.

Comment 7: Line 151. “The fracture of each specimen occurred on the steel side instead of the rivets” Please discuss the reason.

Response to Reviewer #2 comment 7: We have added “This is because the bearing area of both sheets was reduced due to the prefabricated holes. The metallurgical bonding of rivets and Al sheet made up for the shortage, whereas the steel sheet did not.”

Comment 8: Fig. 9. A reason to cause the change in displacement in 3 conditions should be discussed.

Response to Reviewer #2 comment 8: The higher lower the welding speed, the shorter longer the stirring time per unit length of weld, which led to the lower higher temperature, the lower ductility. The sufficient heat input can improve the interfacial metallurgical bonding between 6082-T6 Al and steel. With the decrease of heat input from 100mm/min to 300mm/min, the interfacial metallurgical bonding is weak, which reduces the elongation.

Comment 9: Line 180. “which can be further extended to other dissimilar materials such as Al/Ti, Al/Cu, Al/polymer.” The above sentence is described in conclusion but this point is not discussed in Results and discussion.

Response to Reviewer #2 comment 9: The sentence has been revised as “Since ERFSLW has successfully welded Al alloys and steel, it can be further extended to the welding of other dissimilar materials such as Al/Ti, Al/Cu, Al/polymer, which can be considered as models of “soft/hard”.”

We appreciate for Editor and Reviewers’ warm work earnestly, and hope that the revision will meet with approval. Thanks for your nice work again!

Sincerely yours,

Xiangchen Meng

Round 2

Reviewer 2 Report

Comments and Suggestions for Authors

I have checked the revisions. Please accept the paper in current form. Thank you very much.